# OpenReview forum: "UniGuard: Towards Universal Safety Guardrails for Jailbreak Attacks on Multimodal Large Language Models"
_ICLR.cc/2025/Workshop/BuildingTrust — Submitted to BuildingTrust_

### Official Review · Reviewer_vkfC · 2025-02-26
**The paper presents a defense mechanism to protect MLLMs from jailbreak attacks; however, the experiments should be further refined to provide a more comprehensive evaluation of the proposed method.**

**Rating:** 4
**Confidence:** 5

**Review:**

**Summary**

The paper introduces UNIGUARD, a defense mechanism designed to protect multimodal large language models (MLLMs) from jailbreak attacks that exploit vulnerabilities in these models to produce harmful content. UNIGUARD employs multimodal safety guardrails, optimizing image and text inputs to reduce the likelihood of harmful responses.

**Weakness**

1. The authors evaluate their defense method against a limited set of jailbreak attacks, particularly multimodal ones, such as (1) *Jailbreak in Pieces: Compositional Adversarial Attacks on Multimodal Language Models* and (2) *Visual-RolePlay: Universal Jailbreak Attack on Multimodal Large Language Models via Role-playing Image Character*.

2. When the distance constraint \(\epsilon = 64/255\) is used, it is important to assess whether the image guardrail noise exacerbates the model's hallucination tendencies, potentially leading to undesirable effects.

3. It is recommended that the authors improve the clarity of the paper and Typo errors (line 134). Specifically, in the transferability experiments shown in Figure 5, the proxy model used should be clearly stated.

4. Figure 2, which provides an overview of UniGuard, appears to be missing from the paper.

---

### Official Review · Reviewer_LtcZ · 2025-03-01

**Rating:** 4
**Confidence:** 5

**Review:**

This paper presents a novel defense mechanism that strengthens multimodal large language models (MLLMs) against jailbreak attacks by creating multimodal safety guardrails (image and text) to prevent harmful content. The authors demonstrate its effectiveness across various models, attack types, and modalities, while maintaining the models' vision-language capabilities with minimal performance loss.

Pro: The authors conduct a comprehensive set of experiments across multiple attack strategies and model architectures. The trade-off between model safety and performance is carefully analyzed. Despite the robust defense provided by UniGuard, the impact on benign tasks, such as vision-language understanding, is minimal.

Con: There are formatting issues in the paper, such as Figure 2 not displaying correctly, which affects the reading experience and understanding of the content. Such formatting problems could hinder the presentation of results during the review process, and they need to be addressed promptly.

Based on the experimental results, the simpler Pre-defined Guardrail outperforms the Optimization-based Guardrail in terms of defense performance. However, the paper does not provide an analysis or explanation for this phenomenon. The absence of a discussion on this issue makes the experimental section appear incomplete and lacking depth.

This paper does not demonstrate the effectiveness of the method in defending against structure-based attacks, which are currently a more significant threat in jailbreak scenarios. Given that these attacks pose a greater threat to models, the lack of validation in this area makes the method's applicability in real-world settings seem insufficient.

---

### Decision · Program_Chairs · 2025-03-04

Reject